# Systematics of Forestry Technology for Tracing the Timber Supply Chain

Alexander Kaulen [1,2,*], Lukas Stopfer [2], Kai Lippert [1] and Thomas Purfürst [2]

[1] KWF–German Center for Forest Work and Technology, Spremberger Straße 1, 64823 Groß-Umstadt, Germany; kai.lippert@kwf-online.de
[2] Department of Forest Operations, University of Freiburg, 79098 Freiburg, Germany; lukas.stopfer@outlook.com (L.S.); thomas.purfuerst@foresteng.uni-freiburg.de (T.P.)
[*] Correspondence: alexander.kaulen@kwf-online.de; Tel.: +49-0176-17871327

**Abstract:** Traceability is the ability to follow the processes that a raw material or product goes through. For forestry, this means identifying the wood from the standing tree to the mill entrance and recording all information about the technical (production) and spatial (transportation) manipulation of the timber by linking it to the ID. We reviewed the literature for developments in timber flow traceability. Findings range from disillusionment with the non-application of available forestry technology to enthusiasm for the advancement of technology that—given appropriate incentives of an economic, environmental, consumer-oriented and legislative nature—can rapidly lead to widespread end-to-end media-interruption-free implementation. Based on our research, the solution lies in optical biometric systems that identify the individual piece of wood—without attaching anything—at three crucial points: during assortment at the skid road, at the forest road and at the mill entrance. At all of these points, the data accruing during the timber supply process must be linked to the ID of the piece of wood via data management.

**Keywords:** timber supply chain; traceability; timber identification; data management; technology systematic

## 1. Introduction

Given the positive connotation of wood and wood-related products in society, forestry is gaining interest worldwide. Customer demand for sustainable products drives supply-chain tracing [1]. In forestry, the main driver is illegal logging and deforestation. In Europe, the European Timber Regulation (EUTR) and customer demand for sustainably harvested timber are prompting systems to trace timber flows [2]. Manual and paper-based systems, like branding, plastic tags, forest paint and certificates of origin from the Forest Stewardship Council (FSC) and the Programme for the Endorsement of Forest Certification Schemes (PEFC) still dominate the market [3–5]. With advances in digital technology and higher demands on supply-chain management, automated and digitalised systems are gaining importance [6].

The origin of wood as a raw material will become increasingly important in the near future. The political demands on the forests and the raw materials it provides are contradictory. On the one hand, society's desire for more nature conservation and recreation increases in popularity, spawning demands for decommissioning. On the other hand, growing demands for raw materials conspire with requirements for wood products as $CO_2$ sinks and substitutes for non-renewable substances to incentivize increased forestry productivity [7]. Digital technology, such as current traceability and data processing techniques, can balance sustainability and increased production by managing forests and downstream wood logistics more efficiently, thus ensuring climate-friendly extraction of wood products [8]. Consumers have more power over how products are produced than ever before. Climate-friendly, low-emission and socially sustainable products or raw materials can be marketed through meaningful brands and create so-called intangible

capital through credibility, trust and relevance [9]. Especially since the Kyoto Protocol in 1997, public awareness of the sequestration potential in the Western world has been steadily increasing [10]. In addition to the presumed benefits for the environment, social as well as economic impacts are to be expected.

The newspaper *The Guardian* accuses the current certifications of inaccuracies. FSC and PEFC can only guarantee compliance with their guidelines in the forest. With media discontinuity along the forestry–timber chain, there are security gaps that need to be contained through the use of new technologies [11]. Technical solutions are needed to collect the data required to close these gaps.

Traceability of timber flows through the value chain means creating a link between the raw material and a system like Distributed Ledger Technology (DLT) or an Internet-of-Things (IOT) architecture that holds information about the raw material and the processes it undergoes.

Tracing timber can add value to the product by setting up an information system to detect and locate legal timber and timber products. "Traceability" means the ability to trace any product through all stages of production, processing and distribution [2,12,13]. In essence, a wood flow traceability system enables tracing the raw material one or more steps forward and back at any point in the value chain [14]. Tracing mechanisms are complex due to the composition of individual timber products and timber shipments, which contain a wide range of logs and processed products with different species and sizes. In addition to the traditional methods of stamping and varnishing logs, there are various digital information systems for identifying, tracing and monitoring logs. Automatic identification systems, such as barcodes, QR codes, RFID and microchips, and smart chips, which differ in their practicality and reliability, establish a link between the product, the product database and its process [4,15,16]. Innovative log tracing mechanisms such as DNA fingerprinting are usually very costly and difficult to apply [17]. Older and widely used methods such as colour marking, punching and barcoding still exist and are used effectively in the timber supply chain. Greater efforts towards international cooperation in sharing timber data as part of a global timber traceability information system with unique standards and features promise sustainable supply chains [4,18].

This paper focuses on the state-of-the-art in technology to trace supply chains automatically and electronically. For this purpose, we review different technologies for the identification of wood in the timber supply chain (forest-to-mill) for their suitability, and we present solutions that apply to data management. The technologies are placed in a theoretical overall context.

The aim is **first** to derive a systematic basis of the coverage of current developments in traceability in forestry in the form of a systematic review of the most significant literature, primarily in the period 2011–2022. **Second**, we seek to make a recommendation on how to solve the traceability issue in wood supply.

## 2. Materials and Methods

This review includes academic articles (peer-reviewed), practical guides, technical literature in general and electronic sources. The basis of the search was on the two search platforms for academic articles: Science Direct and Google Scholar. In addition to purely academic articles, Google Scholar also covered articles from and for practice and business. Because of this, the search in Google Scholar had to be limited to the title, while Science Direct queried the title, abstract and keywords. The search string included English and German words, and was used with "OR" and "AND" as Boolean operators in various combinations. Similar sounding words were marked with "*" to make them understandable to the system. The search terms of the object of investigation "forestry", "timber harvesting", "timber supply chain", "unique identification", "forest operation" were combined with the technical terms for each technology class described in section 5. Those included, for example, specific terms like "RFID", "Biometric Fingerprint" or "QR-Code". Based on this, the quick ball system was applied, with which bibliographies or source references/footnotes

were searched for suitable literature. The timespan settings were 2010 to present to keep the results relevant, with exceptions for still-relevant articles. The search yielded many results, few of which had any real relevance. The selection procedure was initially limited to dealing with the title and the abstract. The relevant articles were supplemented with a few recommendations from the literature. The articles remained in their respective categories and were finally analysed intensively in the sense of the scientific objective of this article. The result of the systematic review method yielded the most-relevant and up-to-date articles on the technologies.

## 3. Definitions

### 3.1. Limitations of Consideration

Timber provision includes felling planning and felling preparation, timber harvesting, timber sales and logistics. These are the limits of the technical and spatial framework to be considered. Silvicultural measures prior to wood supply and raw wood processing in the mill are not considered. In the following, we describe a generic wood supply chain for the German-speaking region. Processes vary in detail between regions because of cultural and regulatory differences resulting from the federally organized forest administrations [19]. An important prerequisite is the definition of the boundaries of consideration. The functional unit is the solid cubic metre of wood with bark ($m_3$ over bark) or the batch, which is the selling unit [10,20,21]. The allocation of the functional units to the processes in the supply chain is crucial.

#### 3.1.1. Tracking and Tracing

The spatial and technical manipulation of raw timber can be traced both to the customer and to the supplier of the timber. If it is the former—the tracking of the timber to the customer (downstream)—it is tracking. If it is the latter—the tracing of the supplier (upstream)—it is tracing [22,23].

#### 3.1.2. Planning and Preparation of Felling

On the basis of the annual and natural planning of the forest enterprise, the district manager decides on the cutting sequence, which organises the stand treatment in terms of time and space. The forest manager determines the silvicultural procedure and whether the block formation of several contiguous stands appears to be sensible. Not only annual planning or silvicultural principles play a role, but also the current timber market situation. The forest manager must account for any contractually agreed provision of specific assortments. The marking process and the data collected on felling volume, qualities and assortments already provide important information for timber sales. The work plan contains the forestry operations, the number of forest workers and machines as well as important harvesting information.

#### 3.1.3. Timber Harvesting

Prior to timber harvesting, the forestry enterprise instructs forest workers and machine operators by discussing the work order on site. During the felling, they control workers or are available as contact persons in case of issues. After the timber has been moved to the forest road, the district manager or a forest technician usually records the quantity and assortments of timber with a data recording device. Recording techniques should coincide with those used by the harvester, which serve as control for comparison with the dimensions recorded by the mill. All data flow into the timber list. The timber list contains pile-related timber data and is usually processed with the GPS/GNSS coordinates of the piles into a batch map, which is sent to the timber sales department in the forest enterprise. The batch maps are sent to the customer after the conclusion of a sales contract. In the case of an agreement for free mill delivery, it is sent directly to the timber customer; in the case of an agreement for free forest delivery, it is given to the haulage company.

### 3.1.4. Timber Sales

The timber data in the form of the timber lists are fractured into timber lots by telephone/written form, which concludes the negotiations. If contracts have not already been concluded, such as for long-term purchase of the timber, the contract is now concluded between the sales manager and the timber customer. The latter finally controls the settlement of the invoice. An important step is the timber data reconciliation between the logistic and control measurements taken by the forestry company and the sales measurement, which is usually the factory measurement. The forestry operation compares the measurement records. Discrepancies in quantity and assortment are investigated.

### 3.1.5. Timber Logistics

Free forest delivery is currently the most common method. In the "free forest method", the timber customer commissions a haulage company to transport the timber to the mill. The "free forest method" means that the hauling company or mill takes over the organisation of the timber logistics. From the forestry enterprise's point of view, the work steps are to instruct the driver, if necessary, and removal control.

All processes are depicted here in a highly simplified way and represent the limits of the object of consideration. All processes are meticulously documented, which may facilitate further technical innovations. Harvest planning, marking, harvest control, timber intake and timber transfer consume the majority of the time. These processes hold the greatest potential for optimisation (cf. on this section: [19,24,25]).

### 3.1.6. Functional Unit "Raw Wood"

In order to provide producers and consumers with relevant information, we need to trace the path of the wood from its place of extraction to the mill accurately and free of media breaks. In particular, the origin of the raw material wood and its sustainable and environmentally friendly production and extraction are of special interest to the buyer and customer [20]. Traceability systems (TSs) refer to the recording, storage and transmission of production routes and their underlying data. TSs consist of unique and unambiguously traceable units (TRU). These are clearly identifiable and are occupied with information [12,26]. In the case of the raw wood considered here—the individual log—the volume aggregation (for woodchips) or the pile is the TRU. The information can be quality, strength class, length, carbon storage, harvesting method, coordinates or any other attributes that arise during the manipulation of the wood [27]. Traceability requires a unique ID of the TRU, a data standard for the attached information and a database framework. The latter can be paper-based or electronic, with all information stored on the product ID [20,21]. Probably the best-known paper-based transfer of data along the value chain is certification according to PEFC or FSC. In this simple method, each company passes the certification of the manipulated product on to the next business partner (PEFC/04-01-01 and FSC-STD-40-004) [28]. While the origin of the wood and the presence of a certificate can easily be passed through the chain, the mapping of all relevant wood data is more complex. Here, every spatial and technical manipulation of the wood—be it the harvest itself, the transport, the debarking and/or the varnishing of the end product—is time-stamped so that the entire value chain of the raw wood can be traced at the factory entrance. Where and at what time data are generated during the manipulation of the wood is largely known. However, existing certificates do not adequately depict the traceability of individual processes in the value chain. In order to obtain this information for consumers and business partners, the following prerequisites must be met: (1) uniquely identifiable units (traceable units/TS); (2) hardware that identifies, recognises and assigns information to the units; (3) data transmission without media discontinuity; (4) secure and efficient systems for data storage and management; and (5) higher benefits than additional costs [12,29,30].

### 3.2. Forestry Technology and Traceability in Raw Wood Supply

The object of consideration is located in the field of forestry technology. Erler [31], following the definitions of the term "technology" by the Association of German Engineers, described forestry technology as material systems (artificially created entities) that are used for work in the forest, all facilities and actions in which the material systems are created, and all processes in which the material systems are used. Forestry technology thus includes not only the equipment used, but also the people and the effects of its use on people and the environment. The link between the environment, the people and the technology used is referred to as a process. This is distinct from the working method, which is defined from the perspective of the acting human being. Derived from the science of technology, two main functions of technology can be applied to the part of forestry technology discussed here: manufacturing and transport (Figures 1 and 2). Production classically refers to joining and cutting. Relevant in this work is the cutting that takes place during felling, delimbing, ripping of the top, crosscutting and chipping. It is therefore a (forestry) technical manipulation of the functional unit of raw wood. Spatial manipulation takes place as part of the second main technical function: transport. Transport is the process of moving the functional unit—raw wood—from one place to another. In forestry, the transport of raw wood from the place of felling to the forest road is called "forwarding" if forwarders are used in the CTL system. "Primary transport" is a more general form, which refers to skidding, cable yarding and forwarding. Here, "primary transport" refers to the transport of the wood from the place of felling (and cutting to length, if done at the same place) to a defined place in the stand and to the transport of the timber from the skidding lane to the storage yard along a forest road that is accessible by truck. It is currently carried out exclusively using machines (and rarely by horses). At the storage yard, it is "piled", i.e., stacked into "piles", and separated according to assortment. This process is carried out as an integral part of the primary transport, both in terms of time and technology. Depending on the conditions, other technology may be used, e.g., excavators for clearing storm-drifts, skidders, or cable cranes up to helicopters, especially in the mountains.

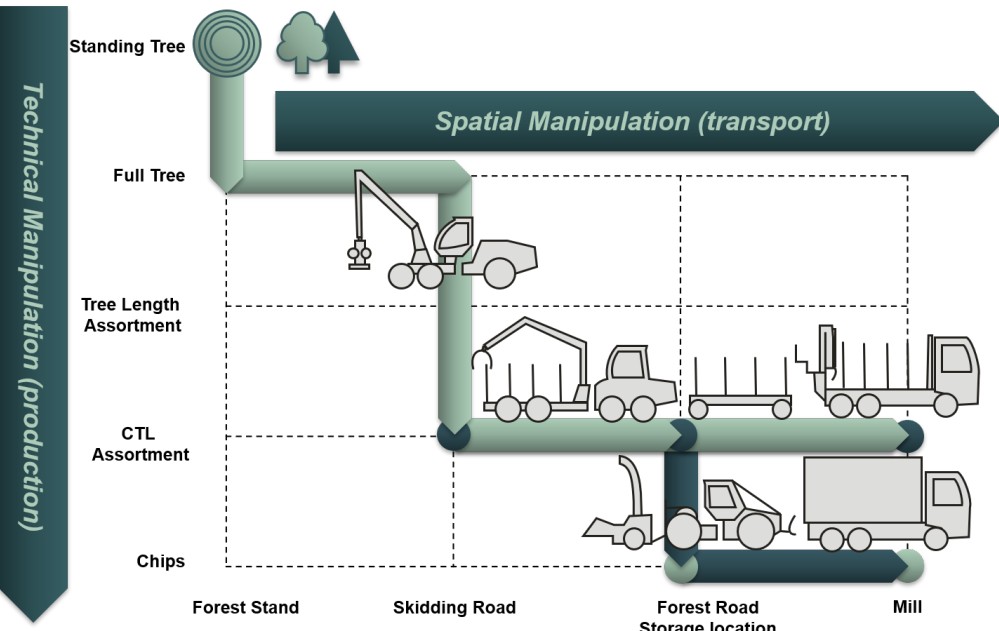

**Figure 1.** In the process of timber supply, the timber goes through several processes. We are talking here about spatial manipulation like transportation and technical manipulation like cutting, which the timber has to go through in order to be further processed in the mill. Here is an example of wood preparation in a highly mechanised timber harvesting system. The process starts with the standing tree, continues with the cutting of the timber in the stand and its advancement to the skid road by the harvester, advancement to the forest road and finally, hauling via truck.

At the timber storage site at the forest road, the presentation of the timber takes place in the course of the sale of timber. During the negotiations between the seller and the buyer, data are often changed (quality, dimensions, and sometimes also the tree species), and individual logs, log sections or piles may be re-sorted and may then have to be relocated. The relocated units are then either assigned to one or more existing lots/batches of the same forest owner or combined into one or more new lots/batches.

The "removal" of the raw wood from the forest to the mill or to another processor requires loading onto a truck. For longer distances, in addition, rail, barge and possibly ocean-going vessels are used [24,31].

The number and extent of manipulations not only generate data but also represent a potential source of error. This requires mechanisms to check and, if necessary, correct the data. In addition, they require dedicated regulations regarding authorizations and data security.

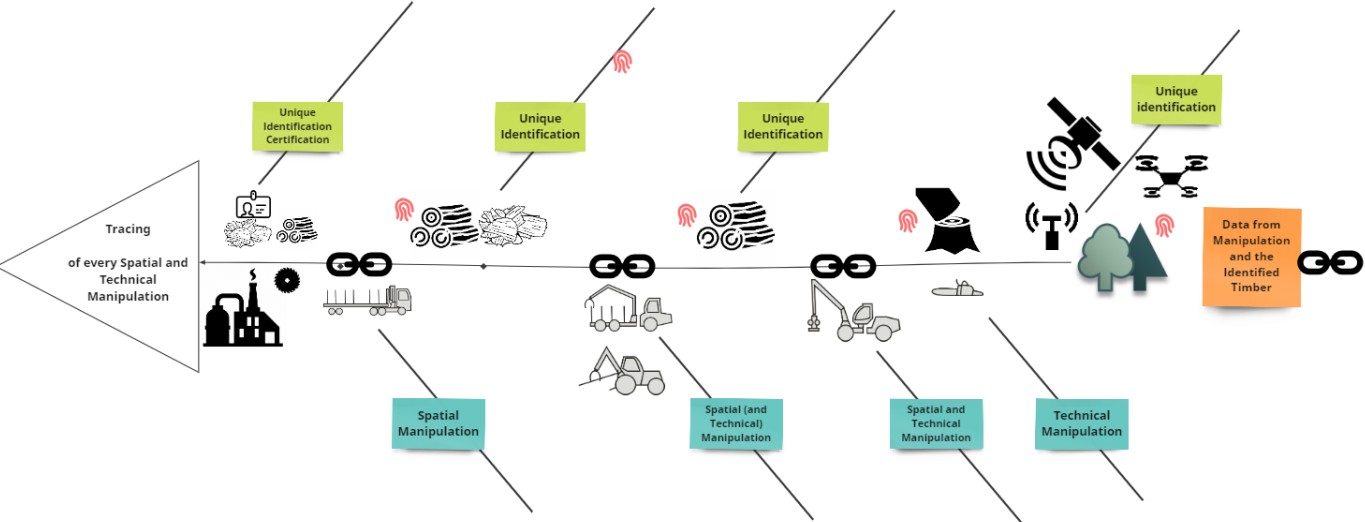

**Figure 2.** In the highly mechanized timber harvesting process presented here as an exemplary case, the technical and spatial manipulation of the timber takes place in order to provide the raw material. During this manipulation, data are generated that contain information about the process but, above all, about the timber itself. In order to preserve this data for later use, they must be "stored" on the functional unit "timber". This requires the identification of the timber at several points in the process: at the standing tree, during the assortment at the skid road, at the forest road and at the mill entrance. The data are therefore linked to the ID of the piece of wood.

## 4. Reviewed Literature

Over the last 25 years, forest science has dealt extensively with the issue of tracing. The main drivers were internal inventory, planning and invoicing. As in many other sectors of the economy, this was mostly still done with paper-based systems. Digital approaches gained momentum in the fight against illegal logging and deforestation. The bulk of traceability systems, and thus also science, deal with in-house technologies and systems. Only a few approaches deal with the supply chain as a whole. These approaches always look at the trading partner with whom one is in direct contact ("one up and one down"). In essence, forestry flow tracing means collecting, maintaining, managing and distributing timber-related data (information on production and properties of the raw material) with the aim of identifying the entities, their history and location along the supply chain in near real-time or retrospectively [32].

The review then lists the most important publications and their respective findings, followed by a summary:

Dykstra et al. [15] looked at standards for data processing, certifications and labelling technologies. They examined chemical paint and chisel labels, branding hammers, conven-

tional labels, nail-based labels, cards, RFID tags, microtaggant tracer paint, and chemical and genetic fingerprinting with the goal of better understanding the chain of custody. In particular, they addressed the advantages and disadvantages.

Schneider [33] describes the importance of marking systems in log logistics. He gives a comprehensive overview of paint markings, chalk markings, plastic plates, RFID tags in all forms, paint on the face applied by the harvester, data standards, QR codes, barcodes and Aztec codes, microtaggants, chemical indicator paint and chemical fingerprints. He embeds the technologies into the processes of forestry and timber management.

Tzoulis and Andreopoulou [34] also address traceability technologies such as punching, paint, barcodes, QR codes, DNA fingerprinting and RFID. They recognise that a globally valid and standardised tracing procedure would be beneficial, although different tracing technologies can perform different tasks.

Mirowski et al. [32] looked at the systems and technologies available at that time for the traceability of timber flows. The requirements for a functional system for traceability are the unambiguous identifiability of the entities or units, the location, the information about the product and the supply process. In most cases, different systems are used for each individual process in the supply chain, and their data output needs to be harmonised. The main drivers of traceability efforts are internal, i.e., economic, and legislative, i.e., legislation. For this, the authors can only recommend RFID tags. For a robust system, the recommendation is less clear but is directed towards standardised data transmission and the associated possibility of merging many systems. Mtibaa et al. [35] additionally present an Internet-of-Things (IOT)-based data model for mapping an RFID-based tracing system.

Dromontt et al. [3] devoted their review to visual, chemical and genetic methods for wood identification with special interest in exploiting synergies in combining different technologies. They dealt with dendrochronology, wood anatomy, mass spectrometry, near-infrared spectroscopy, stable isotopes, radio carbon, DNA barcoding, population genetics and DNA fingerprinting: in summary, using the whole range of forensic methods of wood identification to form a worldwide standardized forensic identification procedure to primarily combat illegal logging.

Appelhanz et al. [20] approached a robust traceability system from the consumer side and their need for transparent information on origin, carbon footprint and similar data. They also considered the additional financial burden. They presented a data architecture model of four layers, which was based on the identification of the wood by RFID tags and QR codes.

Scholz et al. [36] looked at digital technology for supply-chain optimisation with a particular interest in sensor technology. The authors recognise the significant increase in digital technology that is also or exclusively applicable in forestry. RFID tags, GPS-based tracking devices and LIDAR, for example, have been implemented very successfully. At best, wireless sensors such as active RFID technology are suitable for identifying and tracing timber. This publication also deals extensively with the interoperability and integration of already collected, stored and managed (processed) data. One concept that will be discussed in more detail later is the digital twin forest, into which all data flow and which forms a visible and usable platform for timber harvesting.

Godbout et al. [13] recognise the need for a traceability system based on parameters collected for the specific individual tree and later individual log. They note that the key is a database architecture in which all data are referenced to the individual. They also draw attention to the difficulties of such a database.

Fabing [4] gives an overview of the common tracing methods of punching/brand hammer, chemical paint marking, barcodes, QR codes, DNA fingerprinting and RFID tags and provides an outlook on innovative but not field-tested methods such as biometric fingerprinting and remote sensing, and reflectors that announce the location via satellites. He concludes that established manual methods will continue to be used until the new automatic methods become much cheaper and easier to use.

Gasson et al. [37] aimed to offer the consumer a certificate of origin that provides information about the region and species. According to the study, macroscopic and microscopic examinations of the wood, chemical analyses, DNA analyses and various types of spectroscopy are suitable for identifying the species. Statements of regional origin can be made using stable-isotope analysis.

He and Turner [38] as well as Müller et al. [39] provide a larger context between various Industry 4.0 technologies in the forestry and timber industry. Among other things, they also make reference to technologies of tracing in the wood supply chain. For example, He and Turner [38] identify simulation, GIS, RFID, blockchain, IOT, remote sensing data and apps as suitable tools to transition the supply chain to the concept of Industry 4.0. Müller et al. [39] deal with the integration of sensor data, remote sensing and RFID tags into a viable data model of a virtual forest and a virtual timber supply chain.

Keefe et al. [18] recognise the benefits of single-tree collection and monitoring as well as tracking single trees through the value chain. They identify airborne, drone and terrestrial Lidar; RFID tags; production step documentation data from forestry machines; common tracing technologies already mentioned above; blockchain technology; GPS data; common data standards and biometric log identification as drivers. In particular, data utilisation from the forest machines but also the coordinate and biometric fingerprint play an overriding role, especially in the timber supply chain.

Picchi et al. [40], with the goal of demonstrating new methods of wood supply that have less negative impact on the environment, touted the advantages of using remote sensing data, using forest machine data especially from surveying and positioning, and tracing technologies such as paint, hammer branding, coding on the cross-cut section, plastic tags, the biometric fingerprint and RFID tags. In particular, the transferability of data before logging and after mill entry was considered.

Thus, if we look at the scientific starting point of the reviews and works relevant to the topic in an overview, we can state: There is a lot of interest from the scientific community. The majority of the papers deal with RFID tags even though they have not been implemented into practical use on a bigger scale. The technology already exists to meet every demand for traceability quality. However, this also means that the established systems only map fragments of what is possible. The necessary technology to establish a seamless traceability system across the entire value chain seems to be available, although its application must provide a clear benefit compared to the expected costs. This does not yet seem to be the case. Accordingly, a resilient traceability system should be both inexpensive, automatic and designed to cover the entire supply chain in order to offer real added value compared to established, mostly manual systems.

While some publications addressed the issue theoretically, few studies considered the feasibility of the systems and technology throughout the supply chain [32].

Over time, the enthusiasm for RFID tags decreases in favour of optical methods.

## 5. Classification

This section first presents the systematic basis for classifying traceability technologies. In the next step, the main categories within the system are analysed for their viability.

Classification of technologies for traceability of wood flows and especially identification of raw wood are based on the following core distinctions, as as shown graphically in Figure 3.

The first core distinction is **Passive** and **Active**. Active tracing designates the active attachment of a technology to timber. Passive tracing means tracing by the timber itself without attaching identification aids [41].

**Active tracing** is distinguished as using optically readable or radio readable technologies. **Optically readable** includes all technologies wherein a scanner or machine- or human-readable mark is applied to the wood [4,42]. **Radio readable** includes all technologies wherein information can be read from a tag attached to wood via an active or passive signal. The tag either contains information about the wood itself or is used to identify the

wood. In this case, the tag refers to a/the database. The readout takes place via scanners or smartphones [42,43].

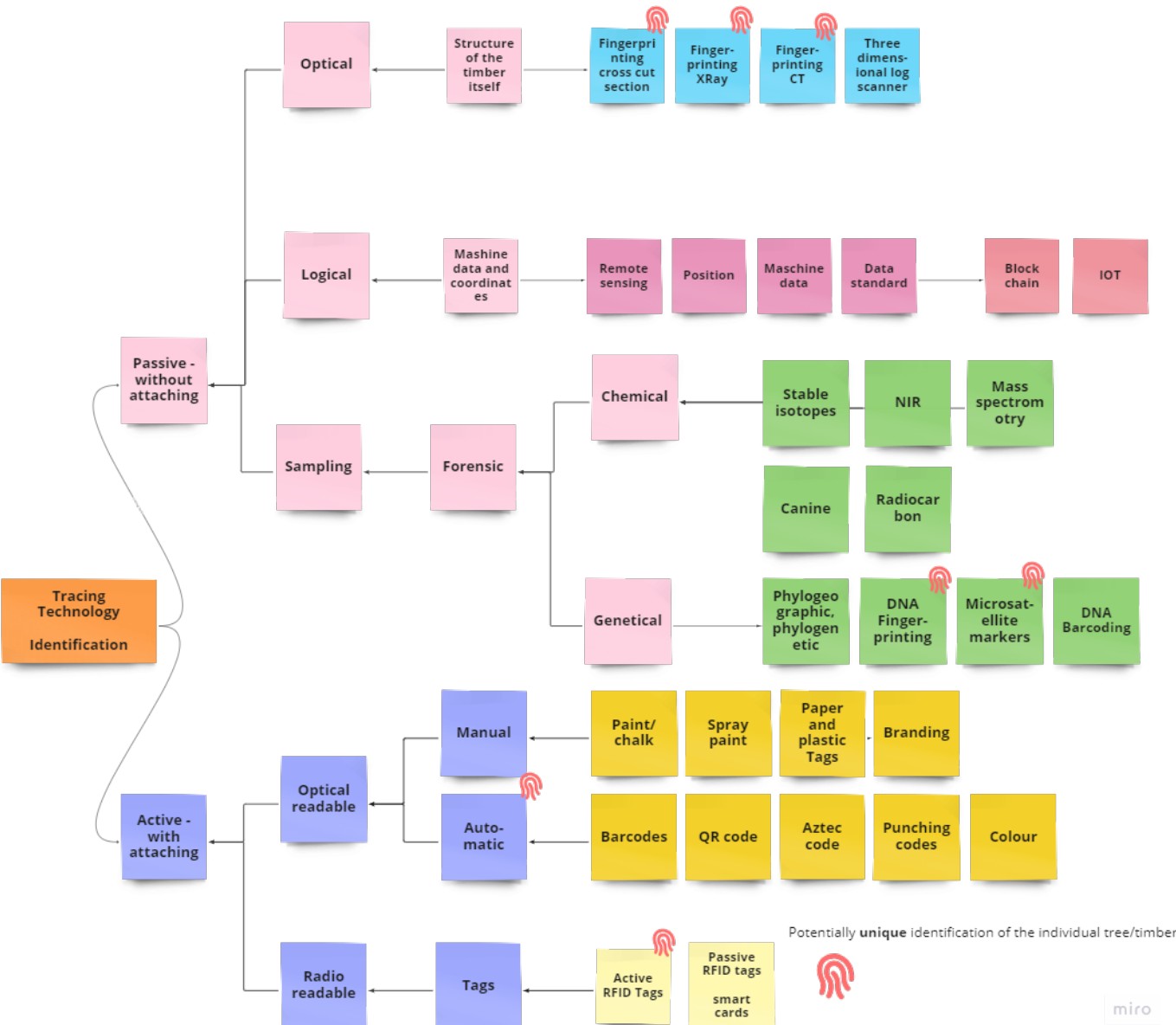

**Figure 3.** Tracing of the individual assorted timber requires identification at several points in the process: at the standing tree, during the assortment at the skid road, at the forest road and at the mill entrance. In order to find out the most efficient method of identification, we assigned each to a system—Passive or Active—and then into Optical, Logical, Sampling, Optically readable, Radio readable and so on.

**Passive tracing** is distinguished as using optical, logical or sampling technologies. **Optical** tracing is passive and is mostly automated, i.e., obligatory procedure in which the structure of the wood is identified by means of computer tomography, X-ray scanner, colour camera or multi-sensor camera [44,45]. **Logical** tracing is the passive accumulation of data generated during technical and spatial manipulation. In most cases, one falls back on automatically recorded machine data and coordinates [46]. These are collected and processed with the machines' own software and are finally transmitted via an interface using one of the common forestry data standards. The evaluation and thus the traceability of the timber flows are handled by enterprise resource planning software. In the simplest

procedure for traceability, this is realised in the passing of certificates [6]. Identical or similar data representing mainly the location and survey results are also collected **manually** on paper via excel tools or digital recording technology [19]. Whether manual or logical, the exchange of standardised data is important [47]. Identification by **sampling** describes the ability to identify the origin, species or even individual at any point in the wood supply chain by taking samples from the wood and analysis and comparison with a database [17,48].

*5.1. Passive Sampling*

Methods for passive-sampling-driven identification of wood include chemical methods such as mass spectrometer examination, infrared spectroscopy, stable isotope determination, and the radiocarbon method; and purely genetic methods such as DNA barcoding, population genetics, and DNA fingerprinting. This is the study of the forensic characteristics of wood, as shown in Table 1 [3].

**Table 1.** Passive sampling class "Chemical". The table shows the five methods of identification that belong to the classification "passive chemical sampling": timber mass spectrometry, near-infrared spectroscopy (NIR), stable isotopes, use of dogs and radiocarbon.

| Passive Sampling, "Chemical" Designation | Viability | Condition | Covered by (Literature) |
|---|---|---|---|
| Mass spectrometry | species, provenance | Raw wood at any stage | [37,49–51] |
| Near-infrared spectroscopy (NIR) | species,provenance | Raw wood at any stage | [49,52–54] |
| canine | limited species | Standing tree to mill | [3,55] |
| Stable isotopes | likelihood species, likelihood provenance | Raw wood at any stage | [49,50,56] |
| Radiocarbon | likelihood species, likelihood provenance | Raw wood at any stage | [3,57] |

The chemical methods, namely mass spectrometry, infrared spectroscopy, canine use, stable isotope analysis and the radiocarbon method are not applicable on their own for solid traceability. However, they are useful and cost-effective methods to randomly check the species, provenance and felling date [49].

**Mass spectrometry** is the collection of a set of molecules that create a chemical fingerprint that is verified with a database [49]. The most common is the ForeST Databank in the United States for the identification of economically important tree species. The DART-TOFMS method allows US authorities to identify species and origin in real time from heartwood splinters [50]. DART-TOFMS uses ions flowing through the sample to create an image of the chemical nature of the sample. The chemical nature varies by species and origin and thus can be used for identification. The main driver is illegal logging [37,51].

**Near-infrared spectroscopy (NIR)** distinguishes between the properties of the wood surface, the chemical composition of the sample, the physical properties in the wood and the moisture content by an infrared signal [52]. In contrast to mass spectroscopy, grinding of the sample, repetition of the spectral analysis and data cleaning by means of calibrations and the deletion of outliers are not necessary. The comparison of the results is also done via a database. The creation of the database is complicated by the lack of standards of the equipment, the software, the settings and the nature of the samples in the procedure [49,53]. The method is particularly suitable for determining the moisture content of wood in the mill [54].

A rather unusual approach is the **use of dogs** that can recognise the chemical composition of wood types. This can only be applied to a few tree species and requires training over at least 5 months [3,55].

**Stable isotopes** occur in nature in varying amounts depending on climate, soil conditions and other external factors. Different ratios in the wood clearly indicate its origin. This is particularly applicable in the identification of illegally harvested timber [50]. In the case of plants, this applies above all to carbon, hydrogen, oxygen and nitrogen. The sample is finely ground and finally subjected to mass spectrometer analysis [56]. The mass spectrometer analysis reveals the amounts of stable isotopes. Both the presence of particularly few heavy isotopes and the presence of particularly many isotopes measured against the standard allow deducing the geographic origin quite accurately with a comparison to the database. The database is usually a map, a so-called isoscape, which shows the probability distribution of heavy isotopes [49].

A variation of stable isotope analysis is the **radiocarbon** method. Here, radioactive carbon molecules (C12, 13 and 14) are analysed for their states of decay [3]. The reference is, sadly, a rapid and short increase of C14 molecules in the 1960s caused by an atomic bomb. Since then, the age of living plants can be determined via radiocarbon [58]. To determine the exact age of the wood, two samples from two different annual rings of the same tree are necessary. The exact felling date can only be determined for raw wood if the youngest annual ring can be clearly identified [57].

**Opportunities:** Passive-sampling-driven identification via chemical components requires only small samples. They usually operate with low financial, time and technical effort. The databases for matching are freely accessible. The preparation of the samples takes little time. Hardly any disposable products are used. Portable devices enable field use. Combining methods yields more-accurate results. The technology and procedures are proven, can be mapped as standard by many laboratories, and are accepted in practice.

**Challenges:** None of the methods has an individual fingerprint as a result. The species and the origin are the output variables. The robustness of the results decreases with the degree of processing of the wood and when multiple species are mixed, when heavily contaminated with chemicals, and/or when exposed to physical processes. The methods require uniform experimental conditions and rigorous testing of the same parts of the wood (heartwood, sapwood, cambium, earlywood, latewood, etc.). Initial investment costs can be high, especially for spectrometers.

**Viability:** Chemical passive sample-based identification is only suitable for determining the species and the origin from a geographically delimited region. This alone cannot result in the unique identification of an individual. However, it serves as a supplement.

**Table 2.** Passive sampling class "genetic": The table shows the four methods of identification of timber—DNA barcoding, microsatellite markers, DNA fingerprinting and phylogeographic/phylogenetic—that are assigned to the classification "passive genetic sampling". The table condenses the information about the viability, condition of the timber to be identified and which literature sources deal with it intensively.

| Passive Sampling "Genetic" Designation | Viability | Condition | Covered by (Literature) |
| --- | --- | --- | --- |
| DNA barcoding | species | Standing tree to mill, old timber, processed timber | [59–61] |
| Microsatellite markers | species, provenance, individual | stump, fresh timber | [62] |
| DNA fingerprinting | individual | Standing tree to mill | [3,13,34,63–67] |
| Phylogeographic, phylogenetic | species, provenance, relationship | Standing tree to mill | [68,69] |

More-promising is the application of genetic methods, as shown in Table 2. For this, DNA from wood samples is extracted, analysed and referenced via a DNA barcode reference database. In general, the method distinguishes two types of genomes that are used for identification: nuclear genome and plastid genome. Nuclear genomes are diploid, are located in the cell nucleus, and change rapidly, such as during reproduction. They are, therefore, well-suited for identification at the individual level. Plastid genomes, on the other hand, are haploid, very stable, unchangeable in the combination of genetic material, i.e., during reproduction, and are located in the organelles. Therefore, they are mainly used for species and origin identification [49]. The range of applications of DNA barcoding extends to the species or family level [60], phylogeographic or phylogeogenetic analysis at the geographic origin level, and DNA fingerprinting at the individual level.

**DNA barcoding** uses short DNA sequences that are extracted from the genome in the cell nucleus to identify the species. The DNA barcode of the individual is compared with a database (e.g., BOLD), which already contains about 60,000 species. The method has the advantage that it can identify species even from highly processed wood, such as paper that has been recycled several times or even timber that is hundreds of years old [59]. It is mainly used to prevent illegal logging in high-value species [61]. The strength of the method of using short DNA sequences and thus being able to operate even with poor DNA quality is, at the same time, its weakness. Species with similar DNA structures can lead to confusion, even if they are rare [3,37,61]. This method is only suitable for the specimen and thus makes an established and cost-effective contribution to identify species but does not contribute substantially to tracing timber through the supply chain [60].

**Microsatellite markers** are fundamentally suitable for tracing wood at the individual level. For this purpose, samples of fresh wood are taken at the felling site and at a point in the wood supply chain. The wood can be identified with over 99% certainty. The technology uses a polymerase chain reaction (PCR) test and must be carried out in the laboratory. The wood must be fresh because the DNA components decay quickly. The method is used for tracing illegally felled wood, especially when there is suspicion regarding illegal logging activity based on a found tree stump [62].

**DNA fingerprinting** identifies genetic markers that vary greatly between individuals of the same species. The comparison with an extensive database makes it possible to recognise the individual. The probability of error is present, although it is low. This method is recognised worldwide, including in other supply chains. Single nucleotide polymorphisms (SNPs) are used. These are variations of single base pairs in the genome that highly specifically distinguish individuals from each other [13]. Single nucleotide polyphormism (SNP) markers are the answer to the short-lived nature of DNA used for microsatellite markers. Thus, traceability can be guaranteed even for degenerated DNA [63,64]. SNPs work on the species, origin and individual levels. They are associated with increased expenditure because of the laboratory activities [65]. DNA fingerprinting is applicable for tracing wood from seed to the final recycling stage [13]. This method is repeatedly recommended for use in the traceability of value chains based on living raw materials [3]. Tracing through the supply chain is possible without additional labelling if samples are taken at key points [66,67].

**Phylogeographic** or phylogenetic describes the analysis of relationships between species but also within species of different provenances [68]. This technique provides information about the species and the origin. It is used elsewhere, for example, when searching for tree species with similar characteristics to exploit new utilisation options. Poor quality of available DNA is a limiting factor here [69].

**Opportunities:** In the fingerprinting method, passive genetic sampling enables the identification of raw wood at suitable points in the supply chain. The other methods are suitable for the identification of species, origin and wood properties and can provide reliable information in case of concrete suspicion of illegal logging [15]. The field is constantly evolving and there is a spirit of optimism that stimulates interest in further development [3,57]. The greatest strength of genetics is the identification of closely related

trees and the precise geographic origin of an individual. The methods are inexpensive. New or unexpected species can be unmistakably identified [34,49].

**Challenges:** A general challenge in all genetic analyses is the poor quality of DNA in heartwood and sapwood. Hardly any intact DNA sequences can be found in them. Furthermore, there are only insufficient reference databases where the sequences of the different species and their provenance are recorded [3]. The intrusion of fungi and other foreign DNA can lead to imprecise results. In addition, the samples, especially for rare and expensive woods, are small and thus contain an insufficient amount of usable DNA [15]. For all methods listed here, a robust database is required to match PCR results. The laboratory tests are time-consuming and expensive. In particular, the methods just described require know-how and are associated with high investment costs, the effort of which cannot be borne by developing countries. The industrialised countries must provide assistance here [3]. The results can be falsified by inbreeding and rapid changes in the DNA, or their interpretation can be made difficult [49,57,67].

**Viability:** Dormontt et al. [3] recommend combining several methods to obtain all the necessary information for forensic investigations. This is because no single method provides all the information. Furthermore, it is recommended that rapid, high-quality field instruments be developed for each site in addition to standard laboratory tests. This reduces inhibitions to use and thus helps to achieve the goal of reliable traceability [37,60]. DNA fingerprinting works on an individual level. Tracing through the supply chain from the standing tree to the entrance of the mill is possible if the sample analysis is carried out at key stages of spatial and technical manipulation: at the tree stump, at the forest road and at the mill entrance [67].

### 5.2. Active Optically Readable

Active optically readable identification of wood includes manual methods and automatic methods, such as the conventional methods of paint, stamp codes, hammer blow, paper tags and plastic tags and their digital successor techniques of barcodes and QR codes, as shown in Table 3. Their main purpose is identification by means of optical features. These can be recognised and re-identified by the human eye or read automatically and are usually checked against a database that links wood-related data with the ID of the piece of wood.

**Table 3.** Active optically readable: The table shows the two core distinctions, manual and automatic, in the active optically readable technique for the identification of timber. The table condenses the information about the viability, condition of the timber to be identified and which literature sources deal with it more intensively.

| Active Optically Readable Identification Designation | Viability | Condition | Covered by (Literature) |
|---|---|---|---|
| manual | any information of individual pile or log directly attached or a number linked to a database | Standing tree to mill, logs, log sections and batches | [4,32–34] |
| automatic | identification via code of individual pile or log linked to a database | Standing tree to mill, logs, log sections and batches | [15,16,32,34,70,71] |

The class of **manual methods** in the active optically readable category includes **paint and chalk markers, spray paint, and paper and plastic tags**. Paint and chalk markings are the first generation of marking and have probably been used for thousands of years. Spray paint facilitated the identification of timber from 1930s onwards. Plastic and paper

tags usually contain an unique identification code that links the wood to data-driven information in a database [4]. Paint and chalk markings are used in a variety of ways. The information applied usually includes the length and diameter, the forest owner and the timber buyer [33]. Developments are also still possible for forest paints. Luminescent nanoparticles are not visible under normal light and only become visible under laser light. This provides additional options for the unique identification of logs, log sections or batches [34]. Likewise, developments can still be seen in traditional punching. While the marking of the face with a hammer blow code is thousands of years old, in modern systems, it is now possible to automatically place the code in the harvester head and have it read automatically in the mill by recognition systems. Codes and information are linked in a database [34]. Number tokens are usually used for single logs or log sections. The specific number can be matched with a table. This enables unique identification. For logs and log sections that are also or exclusively intended for the pulp industry, cellulose quality variants can be used [33,34].

The class of **automatic methods in the active optically readable category** contains **barcodes and QR codes**. Barcodes and related technologies are examples of largely digitised and automated marking. Barcodes and their multidimensional offshoots are suitable as printable markings. They can be applied via laser engraving, in paper form, on plastic or as a colour print directly on the wood. The classic barcode is an established and robust alphanumeric coding in a combination of wide, narrow, black and colourless bars. The coding exerts fixed rules. For example, a start and end character are stored, which define the limits of the information content. The barcode is read by fixed or mobile scanners. The reflection of the different bars from the sequence is evaluated electronically. A disadvantage of the system is its susceptibility to contamination and damage, because as soon as only one bar is damaged, the code becomes unreadable [34,70]. Advantages are the ease of use and the possibility of analogue sighting of the alphanumeric code. Optical character recognition (OCR) consists of a sequence of large and small characters in a matrix and represents a supplement with higher uniqueness. The disadvantages concerning pollution and damage are also given here. After the one-dimensional codes just presented, further improvements are offered by two-dimensional codes, which are displayed via a matrix. The best known forms are the QR code and the Aztec code. They involve the stacking of barcodes. They can be read automatically because the reading device can clearly detect the outline. They naturally contain more information than a barcode [16]. The decisive advantage for wood supply is the presence of redundant data, which can compensate for matrix failure of up to 25% in cases of destruction and contamination without loss of information [16,33,34,72,73]. A special form is the automatic spraying of an individual code through the harvester aggregate. Here, a unique code is printed on the log section for re-identification. Corresponding systems are already in use in Scandinavia [74]. A similar system operates by punching a label via harvester aggregate into the cross-cut section, which can be re-identified later in process, as in the mill [73,75].

**Opportunities** are the cheap, weatherproof and versatile application on all assortments of raw wood. Automatic methods usually require a printer/puncher and a readout device but are unambiguous and tamper-proof. No special training is necessary [32]. QR codes are field-proven, can be used from the batch at the forest road and are characterised by low costs. The codes can be applied automatically to the logs or log sections and can also be read automatically. In principle, they can be used cost-effectively for electronic identification along the supply chain [71].

**Challenges** lie in the time-consuming handling, which burdens labour costs. Manual methods in particular can only be used for large assortments on individual logs or batches or log sections and are difficult to decipher if the writing is unclear. Unfortunately, it is manual methods that are susceptible to counterfeiting. The same applies to hammer branding with the additional poor legibility, which leads to uselessness in identifying single logs. Barcodes are hard to read in difficult conditions, such as dust, and require both a printer and a reader. Plastic tags cannot be made in the forest and must be brought in. If

they are attached with a nail, removal can be a problem. The compostability is only given in the fewest of cases [15,32]. Obviously, actively attached marking systems have a problem with dirt, snow, knots and rough, uneven surfaces [76].

**Viability:** In general, the optical marking and identification technology of QR codes, which are adapted to today's electronic and automated data management needs, is suitable for mapping process transparency and tracking timber flows. QR codes and its relatives have few chances for global application, including wood logistics [33].

### 5.3. Active Radio Readable

**Active radio-readable (radio wave readable) identification** of wood is limited to the numerous methods of **radio frequency identification (RFID)**. RFID tags have established themselves as a resilient application in supply chains. They can be used passively or actively (Table 4). Whereby passive tags respond only to the presence of a scanner. Active tags are battery-powered and allow active transmission of position in real time, allowing permanent tracking. Active tags are used in freight containers [36]. They enable automatic identification through sensors. They are experiencing increasing popularity in business areas such as logistics, invoicing and goods movement in general that require optimisation, cost reduction, and management. However, despite numerous studies and practical trials, they have not been able to establish themselves as a standardised method in wood supply.

**Table 4.** Active radio readable: The table shows the two core distinctions, active RFID and passive RFID, that belong to the classification "active radio readable". The table condenses the information about the viability, condition of the timber to be identified and which literature sources deal with it more intensively.

| Active Radio Readable Identification Designation | Viability | Condition | Covered by (Literature) |
|---|---|---|---|
| Active RFID | active information stored, active positioning | prior harvest standing tree to mill in any condition | [14,15,20,33–36,39,42,72,77–84] |
| Passive RFID | identification via code linked to a database | prior harvest standing tree to mill in any condition | |

Control over the value chain can be mapped via RFID tags by uniquely marking standing trees, raw wood, intermediate and end products. RFID tags are relatively expensive but offer advantages, especially in terms of their resistance to environmental influences and the effects of logging and transport [72]. Tags with a high frequency density can be read over a distance of 4 m and thus form a link between the real unit and the digital image. Unfortunately, tags that are both stable and can be read from a distance exceed the cost of USD 0.41 per solid cubic meter, which represents the upper limit for cost effectiveness [77]. At the stand level, tags can be used to simplify inventories or to target trees for harvesting. RFID tags can hang on the tree for up to a year without interference, which is the period from classic tagging to harvesting. The range of application is mainly in the value timber sector and in large aggregations in order to keep the relative costs low [34]. Fastening with a screw in the tree is recommended to ensure flexibility in growth and environmental conditions as well as reuse. Removing the tags upon receipt at the mill also prevents destruction in the production process. Most mills have strict guidelines regarding plastic contamination of the wood [34,42]. RFID technology is applicable in various ways and processes in wood supply [20]. In most cases, the methods require only a hammer/stapler/screwdriver, nails/screws/staples, the NFC chip and a cellphone with a GPS amplifier. The cellphone transmits all relevant information to the chip. The position is determined automatically. As a result, the break-even point for the RFID method varies

depending on the scope of use. It is more expensive (USD 0.28 per log) but means a time saving of half. The RFID method saves 10%–25% of the costs compared to marking with paint in those areas of wood supply that have a logistical component. The tag can be attached to the tree with a stapler during (negative) marking. The tag must be removed prior to motor-manual felling and reattached after felling. As automation increases, so do the costs (the need for materials and training). The increasing costs are due to the additional costs for materials and know-how. With every technical and spatial manipulation of the wood, the information is stored on the RFID tag and flows into the digital forest model (digital twin) as a change. This method is not yet economically viable, but it is technically feasible. Area-wide application could reduce the financial expenditure significantly [34]. In this context, demand and customer-oriented timber harvesting is recommended [78], combined with the choice of the lowest harvesting costs and the highest expected income by the forest enterprise [84]. In particular, there is a need for special protection of standard tags for use in the forest, regulations for standardised use, standardised interfaces to a unified tracing system, extended raking ranges, and more favourable use ranges [79–81].

**Opportunities:** Modern blockchain technology can be used in combination with RFID technology to create a link from the origin of a tree and the finished product by closing some information gaps. All processing-related information is stored as a product information card belonging to the individual tree or assortment and is made available digitally via a reference web interface. In this context, RFID technology serves as a key technology to link the physical world with the digital world and to create a virtual counterpart. Assessment of the future viability of the technology is considered positive in terms of demographic change and the associated labour shortage due to a high degree of automation [14]. Tags can be read and data stored quickly, while the overall level of security is high. At the same time, the reading reliability is high and the tags can be decoded at any stage of the supply chain [15,39,72]. Potential for the future lies in longer ranges so that data can be read while passing by [14]. Although the tag is applied manually and can theoretically fall off, this has hardly occurred in tests [82]. Compared to other automatic methods such as optical methods (barcodes, paper, etc.), RFID has clear advantages in terms of robustness, reliability and range. Active and passive tags can be used flexibly depending on the intended purpose [36].

**Challenges** include low read range, cost, plastic-based material, low acceptance, a lack of backup in case the tag is destroyed and a lack of standards in the process [14,15,33].

**Viability:** Referring to the timber logistics chain, RFID tags thus allow traceability through the process of tagging, harvesting and logistics to the mill. The tags require an individual code that can be linked to information at any time by a timestamp through a software-based data infrastructure. In the best case, the tag is easily and widely readable, recyclable or unlockable for the paper industry, automatically attachable by a harvester in highly mechanised harvesting, cost effective, weatherproof, shock and vibration tolerant, and, finally, waterproof [72]. RFID does not yet meet these requirements, which has kept it from widespread use. Nevertheless, when widely used, it can map the status of environmental impact, make potential savings visible, enable remote control, and form the basis of environmental compatibility certificates [83]. Remedies could include UHF RFID standard tags according to ISO 18000-6C, whose range covers 100 m and can, in principle, also include sensor technology to determine temperature and humidity. Their use only becomes economically viable with larger volumes or high values [36]. An overall technological package maximizes the exploitation of RFID tag capabilities. This package consists of: a tablet, the Treemetrics Forest app to read inventory data, the RFID tag, an RFID reader/programmer to read and populate the tag with information, a GPS receiver to determine the location of the wood, and a stapler. The conglomerate of technical efforts, together with remote sensing inventory data from a UAV (unmanned aerial vehicle) and a TLS (terrestrial laser scanner), forms a digital twin of the forest in which the trees to be removed are clearly marked [35,84].

### 5.4. Passive Optical

**Passive optical** identification of wood forensics is basically suitable for digitised traceability of wood in the logistics chain, as summarized in Table 5. The automated analysis of the front surface and wood structure inside through machine-assisted recording and processing by algorithms has great potential. The results include species and even the individual. Development has not yet progressed to the point where this process can be applied across-the-board and at the functional unit level [3,37].

**Table 5.** Passive optical: The table shows the four technologies for the identification of timber that are assigned to the classification passive optical: biometric fingerprints by cross-cut, biometric fingerprinting by X-ray/computed tomography, three-dimensional log scanners, and microscopic analysis. The table condenses the information about the viability, condition of the timber to be identified, and which literature sources address it more in depth.

| Passive Optical Identification Designation | Viability | Condition | Covered by (Literature) |
|---|---|---|---|
| Biometric fingerprint by cross-cut | The structure of the timber is its unique ID | first cut to mill in any condition | [3,18,37,40,85–94] |
| Biometric fingerprinting by X-ray/computed tomography | "" | "" | [3,95–97] |
| Three-dimensional log scanners | the surface of the timber is its unique ID | post harvest to mill | [98–100] |
| microscopic analysis | costly and not applicable in field | any stage, even wood chips | [98–100] |

Trees react to their environment and reflect its influence in their growth. Dendrochronology can therefore provide information about past events such as rockfalls, fires, insect infestations, browsing by game, droughts, wind, thinning, lack of light and additional light [85].

Experts distinguish between **macroscopic** and **microscopic** analyses. Microscopic analysis is costly and cannot be automated, while macroscopic analysis can identify the structure of the wood visible to the eye through different instruments at the level of the individual [3]. Such instruments are near-infrared spectrometers and near-infrared hyperspectral cameras for recording cross-cut sections and delimbing sensors for recording irregularities along the log, such as knotholes and decay [86]. Unambiguous tracking of wood flows based on unique images of the cross-cut section can also be called **biometric fingerprinting** [40]. Biometric fingerprinting for tracking wood is the visual and much cheaper counterpart to tracking systems that require active application of information technology such as QR codes or RFID tags [87,88]. The combination of visible and invisible (to the human eye) spectral analysis is expected to significantly increase the likelihood of unique identification and re-identification. Machine learning is gaining importance based on the data collected with this technology [89]. In simple terms, the identification of the individual cross-cut section takes place via three surveys: the spacing and shape of the annual rings, the shape of the outline, and the structure of the wood itself result in an unmistakable fingerprint. Pattern matching is done by a database and software; the process can be repeated at any stage of the wood supply, and thus, the wood can be re-identified [90,91,101]. The method of matching fingerprints is template matching, in which the initial image of the wood is compared with images in a database. The pair of images with the highest value of matching is most likely the same piece of wood [92]. For optimal images of the timber, the cut should be smooth (but does not need to be sanded) and free from contamination, and there should be no light coming in from the background.

Optimal conditions are created by taking pictures in the mill or under the influence of a flashlight [93]. In any case, study results predict a reliability of 100% [18,90,101]. A promising experiment aims to do just that. "DiGeBaSt" wants to create fingerprints of the end faces at three points of wood supply and match them with a cloud-based database. The three points of wood supply are: after felling with a camera system attached to the harvester head, a hand-held camera system for images at the woodpile, and finally a stationary system at mill entry. The risk of image blurring due to contamination and wood alteration is to be countered by quickly folding the camera in and out of the harvester head and by the possibility of making a cleaning cut in the mill, as well as by improved software that eliminates environmental and wood-related blurring [94]. Taigatech AB of Sweden is already working with a product available on the market, which starts at the forest road, i.e., with the wood logistics. This eliminates the difficult situation of biometric recording at harvesting [102].

Despite technological advances, tree ring detection is mostly limited to two-dimensional methods and in some cases is still performed manually. The boundaries of the annual rings can be partially wedged into each other, interrupted or only indistinctly recognizable over the entire log, which makes annual ring analysis a special challenge. When using image-based 2D measurement techniques, the cut surfaces must be cleared of unevenness to increase the contrast between earlywood and latewood. Automatic recognition of the three-dimensional structure and width of tree rings from **computed tomography** data is already technically possible and tackles these issues [3]. This approach relies on a modified Canny edge detection algorithm that is capable of detecting all tree rings in the entire image stack. An advantage of this method is that it requires very little to no user interaction. Tree ring boundaries can be partially wedged, interrupted, or simply indistinct across the entire log, making tree ring analysis particularly challenging. The tree ring widths have been calculated using a new algorithm that determines the tree ring widths from the averaged distances between two consecutive tree rings [95]. It was found that the tree ring widths determined by the automatic method were consistent with the manually measured tree ring widths for all samples studied, ensuring the applicability of this method. In addition, the methods also automatically analyse the complete 3D morphology of the tree ring, which helps to better analyse changes to the tree rings. Therefore, time-consuming preparation (e.g., ensuring that tree ring edges are perpendicular to the cut surface) is not required, which reduces uncertainty for less-experienced users [96,97].

**Three-dimensional log scanners** work similarly by creating a fingerprint from the surface of a log with and without bark. The problem lies in the presence or absence of the bark. The surface texture of the log can change significantly in the deployment process [98–100].

**Opportunities:** The procedures are inexpensive and do not require active attachment; therefore, no material is left behind on the log, and identification at the level of the individual can be traced in any technical and spatial manipulation. In particular, the opportunity lies in the automatic standardised identification and the associated possibility to encourage machine learning [37,40].

**Challenges:** Although identification at the individual level is possible through dendrochronological recording, it is limited to tree species of non-tropical origin. This is because the method is based on the presence of annual rings [3]. All optical processes are exposed to influences such as pollution, light quality and vibrations during manipulation [86,94]. The amount of data to be processed for high-resolution images is high. However, the amount of data can be reduced to the key vectors [40]. The wood, and thus the cross-cut section, change greatly during spatial manipulation and are susceptible to contamination. The use of many spectra and wavelengths (visible and non-visible) is therefore recommended [89]. Vibrations during recording can negatively influence the results [100].

**Viability:** The great opportunity for the methods of passive optical identification of the timber lies in the linkage with other sensors that record the nature of the wood in a central database that not only makes the individual uniquely identifiable at each step of spatial and technical manipulation but also generates a digital image of the raw wood [86].

A central issue is the connection between information about the wood that is created during spatial and technical manipulation and the identity of the wood [103,104]. The recordings are already possible by sensors and/or cameras on the harvester, so clear identification of the individual can be established through optical identification directly during the timber harvest [18,86].

*5.5. Passive Logical and Data-Driven*

Three decisive factors can be identified in the **passive logical identification and tracing** of timber, as listed in Table 6: the possibility to already identify and measure the standing tree via remote sensing, determination of the position of the standing tree, identifying a single log or log piece at critical points of the timber supply chain, and finally, the processing of data that are collected with forestry machines (or also manually). In the latter, the standardised communication of data via data standards plays a crucial role.

**Table 6.** Passive logical processes: The table shows four of the many methods that can be classified as based on a passive logical process for identification of timber: remote sensing, positioning, machine data, and data standards. The table condenses the information about the viability, condition of the timber to be identified and which literature sources address it more in depth.

| Passive-Logical-Process-Based Identification Designation | Viability | Condition | Covered by (Literature) |
|---|---|---|---|
| Remote sensing | Position, ID, properties | prior harvest standing tree to harvest operation | [18,105–110] |
| Positioning | Position of the individual | standing tree to mill in any condition | [111–119] |
| Machine data | position, measurement data, production data at the level of the individual | from standing tree to forest road (harvester, forwarder) and beyond (logistics) | [18,109,120] |
| Data standards | standardised input, processing and communication of data from forestry operations | from standing tree to mill | [19,109,121–140] |

**Remote sensing** of the position and attributes of the standing tree offers opportunities for timber supply and forest management. Not only does it represent the first step in tracing timber flows to the mill and the associated ability to interconnect machines that collect data on the functional unit of timber, but it optimises planning, harvesting, primary transport, machine navigation, automation and forest management by requiring fewer personnel and resources to spend less time on each process [18,110]. Remote sensing to identify the standing tree comes down to laser imaging detection and ranging (Lidar). Terrestrial Lidar, drone Lidar and airborne Lidar collectively provide information about the standing tree by exposing point clouds of the reflected lasers to sophisticated algorithms [105,106]. With an accuracy of about 80 cm, the positions of individual trees can be determined and assigned the attributes mean height, mean DBH, volume, aboveground biomass, canopy extent, forest levels, branch texture, tree morphology and tree age [18,107–109,141,142].

**Global navigation satellite systems (GNSSs)** are satellite-based radio navigation systems that provide three-dimensional position determination with time stamps. GPS-enabled hardware installed in mobile devices allows positioning of the user. Without access to the internet, the current position can be displayed on a map. The availability and quality of the GPS signal depends on the installed hardware, location, foliage and weather [113]. At least four satellites are required for this, whereas an increasing number of satellites

available enables higher accuracy. Positions can be determined even more accurately with differentiated GPS (DGPS), where reference points minimise possible errors [111,119]. Further improvements can be achieved by using special antennas such as the choker ring antenna [118]. Positioning by means of GNSS using systems such as GPS, GLONASS, Galileo and Beidou offers a cheap and functional method compared to classical terrestrial surveying methods, but experience has shown that it reaches its limits of usability in forests due to fluctuating and sometimes unsatisfactory precision [112]. The need for accurate position determination in the centimeter and sub-meter range is a critical issue in precision forestry as well as in tracing timber flows. However, research shows that current GNSS receivers operate in the sub-meter range. For example, GIS-class receivers achieve an accuracy of about 1.3 m, and geodetic class receivers achieve an accuracy well below one meter deviation at about 80 cm [119]. When using hybrid techniques like rapid static and total static, even sub-centimeter accuracy can be achieved [113,142].

Discrepancies between the accuracy of data provided by the equipment manufacturers and service providers and experience with GNSS measurements in the forest are sometimes considerable due to the lack of standardisation. In addition to the qualitative differences in accuracy, a clear influence of the location of the GNSS equipment on the vehicle as well as significant dependencies of the accuracy on the stocking can be observed [114]. Tests found that measured distances were 9%–28% longer than the normal lines when driving in a straight line through the forest. The deviation varied from 0 m to 31 m depending on the degree of canopy cover [115]. The degree of canopy cover, the type of canopy tree species, forest types, water content of leaves and branches, topography, temperature, wind, snow, ice and some other site-dependent factors are critical in forest position determinations [112,118]. Nevertheless, portable, smartphone-based and amplifier-equipped systems are making progress and delivering promising accuracies [116,118]. The achievable accuracies in the forest contrast with multiple expenses and uncertainties. On the one hand, the necessary forest-suitable hardware in the form of robust notebooks or tablets with sufficient battery power has not been available or has only been available at unreasonable prices. On the other hand, the attainable position accuracies in the forest have often been insufficient and/or not in relation to the attainable monetary advantage [117]. In general, devices for professional use offer advantages over those for private use. The more expensive and technically advanced, the better. Although accuracy has progressed enormously, there is still potential [118,119].

As described above, **forestry machines** provide **data** on single logs and assortments. For passive logical identification, we take the measurement data of the length, diameter at several points of the log, the volume to be derived from it and the unambiguous determination of the position of the machine, its travel paths and the position of the wood in the stand and finally at the forest road [120]. Also of interest are machine data itself, like production data, performance, diesel consumption and costs (Figure 4). Machine learning and Big-Data processing play an overriding role here, which will intensify in the next 10 years [18,109]. Three-dimensional laser scanners and hyperspectral cameras are advanced technologies, although rollers provide the most data for length and diameter. It is important to calibrate the devices daily and to verify the accuracy of the sensor technology [120,134,135].

The data packets described above are useless if they are not collected, processed and sent in a standardised way. Various **data standards** play an essential role here. Data standards give the digital "information packages" to be exchanged a practical structure and enable interested market participants to process wood-related information uniformly, quickly and securely. The improved networking of the wood supplier, transporter and buyer offers improved digital business processes, higher operational and resource efficiency and, as a consequence, higher contributions to environmental protection. Standards in general and data standards only benefit the forestry industry if they are applied across the board [143,144]. They structure information, ensure error-free exchange, guarantee interoperability of data and are the basis for the use of applications for "Big Data", analysis and prediction [137]. On the one hand, digital applications benefit from the use of data

standards. But on the other hand, we need digital applications that display and utilise data from data standards in a practical way. Lack of data standardisation is one of the most important cost drivers for companies [145]. From Baumann et al. [137], a cost saving of EUR 7–14 per solid cubic meter on average emerges [19]. In addition to the benefits, barriers include the costs to standard adoption, reduced flexibility, reduced market competition and higher prices due to monopolisation [146]. Data standards can have a market-opening effect but also a market-closing effect [139,147].

We identified seven relevant standards with different coverages of the process chain. StanForD covers timber harvest and logisitcs; ELDATsmart covers logistics and billing (the same applies to FHPDAT); papiNET covers ordering, timber harvest, logistics, timber processing, billing and further processing; eFids covers timber harvest, logistics, timber processing and billing; GeoDat only covers logistics; and Forestand covers the forest stand and order placement [121,122]. ELDATsmart/FHPDAT, papiNet and StanForD are described in more detail below. DRMdat, the newest attempt to implement an industry-wide data standard, is also covered.

The **ELDAT standard** was published in 2002 [123]. Initially, ElDat was developed in German according to the principle of the shopping cart in XML format. However, this application principle was interpreted as a disadvantage for the standard, which is why a project was launched in Kuratorium für Waldarbeit und Forsttechnik e.V. in 2015 to advance the standard's unification [124]. In 2018, the new **ELDATsmart** data standard in JSON file format was recommended by the industry [125]. The modules are "timber provision", "transport order", "delivery bill", "measurement protocol" and "billing". Both ElDat and ELDATsmart are open data standards [127,136].

Comparable to the German ElDat standard, an XML data standard for the Austrian forest-timber logistics chain was published in Austria by the coordination and communication platform Forst Holz Papier (FHP) [126]. Thus, while ElDat enables information transfer from the forest owner to the timber buyer, **FHPDAT** is limited to information transfer from the mill to the forest [136]. The FHPDAT data standard is defined as an open standard, but it has severe limitations [127]. The data standard for **Digital Resource Management in Central Europe (DRMdat)** is the most-recent of the forestry data standards. It was developed in cooperation between German and Austrian partners. The goal was to develop a data standard for Germany and Austria—which are closely intertwined in the forestry sector—based on experience gained from existing data standards in the two countries. DRMdat, like ELDATsmart, has been implemented in JSON file format and is also built on basic modules composed of different containers. By using as much mandatory information and as many reference tables as possible, the data standard is intended to be as rigorous in its application as its predecessors [128,140].

The **papiNet standard**, which is widely used internationally at present, was developed as an XML standard in Germany in 2000 at the behest of the European and North American paper industry and printers. Only later was it possible to integrate the supplier side into the business processes [129]. The papiNet standard follows a modular principle, which releases various extensions for use starting from a master document [130,136]. The papiNet standard covers most areas of electronic data exchange in the forestry-wood logistics chain [129]. PapiNet is an industry standard and not an open standard. It was developed by the industry and is supported and will be further developed. Since 2021, the papiNet standard has provided an API specification [130].

**StanForD (Standard for Forest machine Data and Communication)** is the first data standard in forestry. It is limited to fully mechanised harvesting. The most recent version is StanForD2010. With the StanForD standard, it was possible for the first time to collect machine data and production data uniformly via the CAN connections of large forestry machines, in particular harvesters and forwarders, to store them in a standardised way, and finally, to process and send them electronically in a standardised way [131]. Currently, communication (M2M) can be handled via the XML data standard. This includes the transfer of work orders and assortment specifications to the machine, work progress

messages from the machine, and complete production lists including GPS coordinates of the provided wood [132]. Since all stakeholders participate in the development and support of the data standard, it is used by large forestry machine manufacturers [134]. StanForD 2010 is a de facto standard. While it has not been developed and disseminated by a standards organisation, it is used, revised and supported by all stakeholders in highly and fully mechanised timber harvesting [131,133–135].

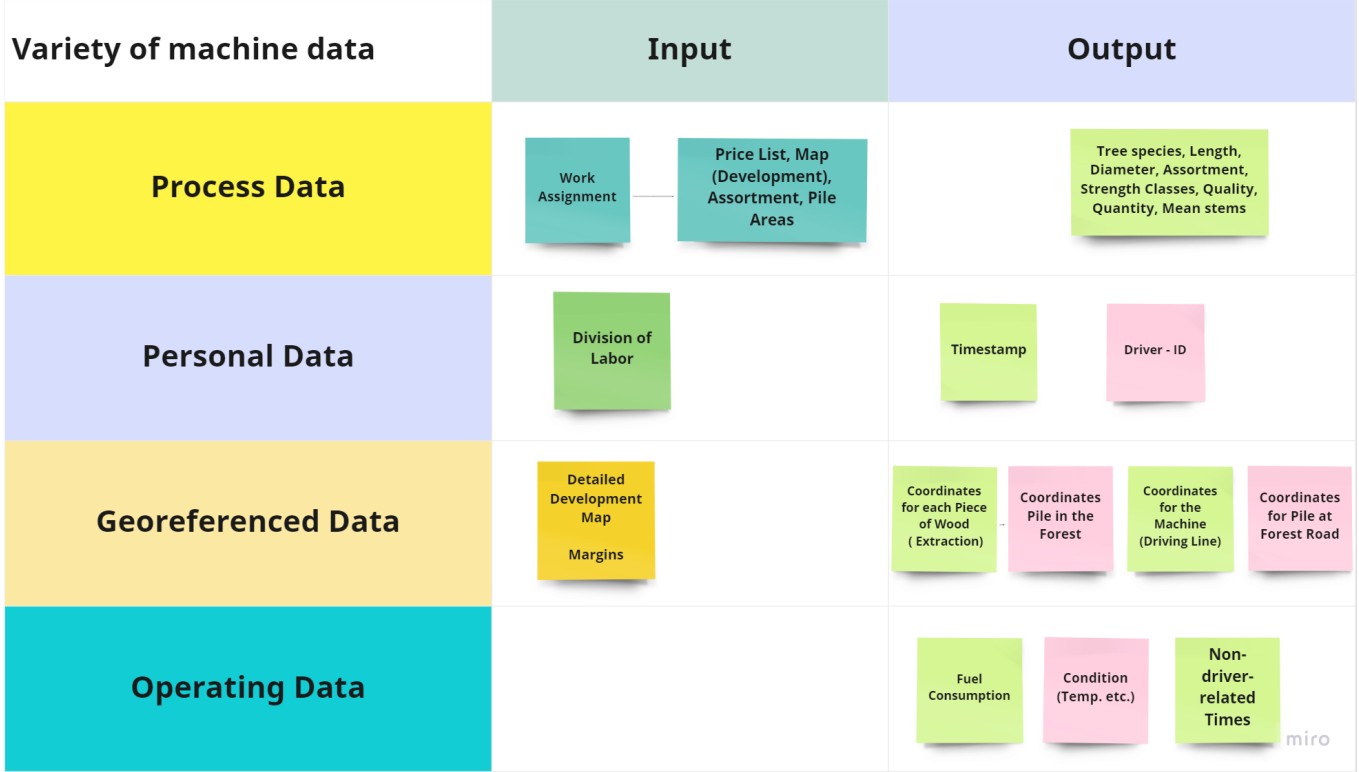

**Figure 4.** Using the example of the data packages supported by StanForD, it is obvious that the designation "data" usually includes many specific data classes. Data generated solely by forest machines are heterogeneous, and embedding them in a media-break-free exchange along the forest-timber chain presents some challenges.

The ELDAT and FHPDAT data standards are national standards. StandForD and papiNet take a more international approach, not least because of the user interface in English. However, there is a risk that country-specific conditions are not adequately represented or that smaller companies that exclusively serve the national market are left out. National standards have more process flexibility [136,137]. In many cases, StanForD data are used for inventories, production adjustments and scientific studies. Linking data to post-logging processes remains elusive [109]. Adherence to data standards yields a cost reduction of EUR 4–7 per cubic meter [138]. The additional time expenditure of 9 min per 100 solid cubic meters speaks against this, according to [19]. With the intensification of international business relations, the pressure to increase efficiency by reducing costs is increasing. In addition, there is an increased interest in smooth communication between companies operating in a market economy. In particular, however, the benefit to the industry as a whole of using a uniform electronic data standard must be demonstrated in order to eclipse operational competitive advantages over other companies [139].

**Opportunities** of the passive logical and data-driven methods to track wood flows are that they use available data, workers are rarely confronted with collection and evaluation, the level of automation is high and the method is favoured by machine learning. Nothing needs to be attached directly to the wood. The methods are accurate. Simply speaking,

many sensors on different machines collect data, the conglomeration of which results in a unique identification.

**Challenges** lie in the high degree of mechanisation. Not every forestry operation or forestry contractor can make use of remote sensing. Older machines operate with a lower degree of automated generation of data. Although timber tracing is considered a by-product of the already-collected data, the cost of modern sensor technology is very high. So while tracing the wood is costless, the initial cost is enormous. In addition, the same level of mechanisation is required at every point in the wood supply chain to ensure that tracing does not break down. The biggest challenges are in data processing, storing and communication, although data standards have already made a contribution to their specific area of application. Assigning passive logical data from harvester protocols to the traceable individual log is difficult due to the heterogeneous data situation. This is because there are data for different qualities and values and different tree species and assortments that go to many different buyers with different demands.

The **viability** of the passive logical and data-driven method lies, as long as the data stream does not break off, between the seedling and the mill entrance. In addition, a problem to be solved is to link the heterogeneous data, which also have a certain volume, with the individual log. A combined system of biometric fingerprinting systems as described in Section 5.4, remote sensing data, logging data from the forest machines and a database structure provides the solution to link the automatic identification of individual logs with the measurement data at the individual log level.

The following section discusses how the required data management can be structured in a target-oriented manner.

*5.6. The Challenges of Data Handling*

Since much of the data collected are lost along the supply chain, a digital, automated approach must be considered. The presented tracing solutions are able to automatically trace timber. Two possible technological solutions are listed in Table 7. For connecting real entities or processes of interest with all the measurable characteristics in real time to a virtual image (digital twin), a data model is needed that can display data of the wood supply chain. The data collected and passed on in the various processes and stations along a multi-layered supply chain and reflecting the nature (the complexity of the properties) of the raw material are subject to factual, spatial and temporal changes [148].

Basically, in a tracing system to map timber flows, the functional unit—the process of spatial and technical manipulation and surveying, i.e., collection of the incoming data— must be brought together [149]. In doing so, the system should also identify and display the position of each individual functional unit at any given time. The system is effective if it recognizes each individual functional unit, provides it with a timestamp and the coordinates of its position, and precisely assigns all measurement data collected with respect to this functional unit to it and stores it on it. Data collection performed at several stations of the process is especially necessary to document all technical, spatial and temporal changes caused by the different manipulations [131,150]. Decisive in the system is the relationship of the functional unit to its state before and after the respective technical or spatial manipulation. A parent ID and a child ID clarify this relationship. In the data model, all data about the respective functional unit are assigned exactly to this functional unit and are stored on it during the entire duration of its existence [151].

**Table 7.** Data handling: The table shows two of many methods to identify timber via distributed ledger (blockchain) and IOT/digital twin, which are classified as based on passive logical processes. The table condenses the information about the viability, condition of the timber to be identified and which literature sources address it more in depth.

| Data Handling Designation | Viability | Condition | Covered by (Literature) |
| --- | --- | --- | --- |
| Distributed ledger (blockchain) | one block per technical or spatial manipulation | data from every step of the wood supply chain | [14,152–157] |
| IOT/digital twin | data-driven digital counterpart to the real asset | entire supply chain | [39,141,149,158–160] |

A **distributed ledger (blockchain)** is a continuously expandable list of records. In addition to its classic areas of application in cryptocurrencies such as Bitcoin and Ethereum, a blockchain in connection with smart contracts (mutually agreeable conclusion of an agreement) is predestined for solutions that require more traceability and data security in supply chains. The data records are understood as blocks that are connected to each other as a chain of individuals with the help of an encryption algorithm [161]. The blockchain thus represents data infrastructure with which the exchange of goods, commodities and information can be mapped. During a transaction, the information is first exchanged between the action partners, and then the entire database is synchronised with all the participating actors (nodes) provided that nothing prevents the transaction. Thus, the blockchain is a decentralised database. The distributed information change is stored in identical form at each participant of the blockchain, so changes require the consent of all partners [152]. Each partner has insight into the same sequence of blocks. Therefore, each partner has a backup. Several stakeholders and shareholders are provided with access to the same information depending on the type and rules of the blockchain [153]. Forgeries are immediately visible [155]. In terms of continuous traceability, all blocks are traceable back to their origin, and so are changes to blocks. Thus, the blockchain model is designed to provide a high degree of transparency. Manipulation is largely ruled out. For example, users are excluded from the network if fraud is suspected. The personal rights of users are guaranteed through anonymisation by assigning each one a user ID [154]. However, if the user ID or password is lost, the data records can no longer be accessed. Blockchain technology thus offers decisive advantages that prove useful for traceability [14]. The application of blockchain can ensure transparency across all processes in a value chain. This leads to considerable increases in efficiency and forestry digitalisation with simultaneous cost savings, time savings and improved planning reliability [155–157]. More specifically, the distributed ledger (architecture) applies the unique identification of individual logs (tracking) with the information (wood-related data) about the individual logs via electronic (automated) input [14].

**Opportunities** are expressed in the exclusion of replacing or changing data without the consent of the partners involved. Participation in the blockchain and the generation of new blocks requires the fulfilment of smart contracts [154]. The blockchain simultaneously creates privacy and transparency, as everyone has access to the data but the actors only appear in code. Tracking is accurate and seamless because no new block can be generated without a relationship to the previous blocks. This helps fight against abuse, corruption and, especially, illegal logging because every area where timber can be legally cut can be marked. Unauthorised trespassing immediately results in declaration as illegally logged timber. Acceptance of this timber is prohibited. This means that the basic requirement for a contract with the smart contracts is not fulfilled. Certification of wood can be done directly through the uniquely identifiable origin. Running the system is costly and energy intensive but can be done in a decentralised manner: for example, in remote areas that have low electricity prices. Mills could do their own mining and use the waste heat for drying. A

blockchain also promises cost savings. Forestry companies and sawyers no longer have to rely on banks, IT service providers and similar services. Transactions are handled directly via smart contracts without a third party. In addition, there is no longer a need for central servers, which reduces costs and eliminates data loss [155,157]. A blockchain makes forest owners independent of certifiers [152] even though the programme for the Endorsement of Forest Certification Schemes (PEFC) is working on a blockchain solution [28].

**Challenges:** Blockchain technology is still in its infancy with respect to global supply chains. Forestry and timber companies are not willing to take the risk of investing. If the user ID and password to the wallet are lost, access to the data is also lost. The independence from institutions is also a disadvantage. This is because standards and rules by official bodies do not exist. The question of the legal resilience of smart contracts must be raised [154]. Thus, institutions cannot intervene in transactions in the case of criminal backgrounds. Service and administrative positions are at stake [155,157].

**Viability:** Blockchain or other distributed ledger technologies are suitable for mapping supply chains. Each technical and spatial manipulation is represented by a block in the chain. All essential information is stored in the block. The connection between the blocks is a unique value calculated by an algorithm.

Studies recommend the combination of **digital twins** and blockchain to ensure real-time management of supply chains. Blockchain can provide the needed security and transparency. The digital twin organises collaboration in a blockchain and better maps processes [158]. It is the digital image of physical objects that can communicate with each other within a cyberphysical system. Physical objects can be anything that is relevant in the supply chain. For example, digital simulations can be created of trees, soil, harvesters, foresters, harvesters, timber and trails that communicate with each other in the digital space [159]. The connection of wood, data packages, hardware and software, stakeholders and the working human in forestry through the **Internet of Things (IoT)** creates a space by corresponding, planning and optimising [141,149].

Interfaces regulate the communication between individual machines and applications by authenticating participants to authorise communication through the identity provider and by establishing communication paths through the directory: for example, regulating data ownership and rights. Data storage is cloud- or edge-based. The cloud is appropriate when dealing with non-machine digital twins, such as humans or wood [149]. Using all available data from the timber harvesting process, the technology allows the construction of a digital twin that harvesting machines can use to orient and navigate in the real forest [160]. The machines can—at least theoretically and possible technically—capture all relevant data of the raw material, and at the same time, the IOT creates a virtual image of the environment in the sense of an automated partial inventory of the stands [39].

**Opportunities:** A digital twin can mean significant time savings and overall increased efficiency in data-driven communication. The cyberphysical system solves the main challenges of networks in the forest: the lack of internet, data security and data ownership. Cyberphysical systems operate in a decentralized manner and do not rely on servers and cloud access. The standards of the structure of a data model are already elaborated by the industry-driven models [162]. The opportunity lies in the real-time dissemination of data and their linkage, which finds new indicators to improve value creation. IOTs are characterized by low energy costs and low data volumes with simultaneous fast data transport and data processing [159]. Especially in the forest, devices can continue to run even without connection to the Internet. After the connection to the cloud is restored, the generated data are then updated [149].

**Challenges:** Not every physical asset can be networked, so valuable information is lost. A lack of standardisation of the data packages to be exchanged can lead to problems with the interfaces. Clarity must be established about the ownership of the data to be administered [159].

**Viability:** IOT can map supply chains by connecting the individual machines and protagonists and generating a digital image of the real assets but also of the process flows.

Nevertheless, the digital twin is a bit too oversized and overambitious for the simple mapping of wood flows. But the derivation of wood flows can be a by-product of a forestry-timber digital twin.

## 6. Discussion

The key to traceability is the unique identification of individual logs, log sections, lots and batches without media breaks along the entire value chain.

**Complexity:** The major challenges lie in the complexity of the value chain. Especially in the German forestry sector, heterogeneous stakeholder groups enter into decentralized business relationships with each other. Each technical or spatial manipulation of logs runs independently with specialised machines and applications. Although this network of processes is obviously interconnected, no linear data communication can be detected. It can take up to 12 weeks for simple messages to be transmitted between timber buyers and sellers. This makes agile responses to changing circumstances difficult. Additionally, there are not only demands for the provision of the raw material but also those of stakeholders from conservation, tourism, professional associations, water management and many more [162]. The multitude of participants in the value chain, the complexity of processes and the lack of linear networking of data flows can also be understood as an opportunity in the field of digital twinning [149]. Timber harvesting and logistics are subject to natural events such as weather conditions but also extreme events such as bark beetle infestation, wind throw and snow breakage. Planning reliability suffers as a result. It is all the more important to support flexibility with digital solutions [162]. On average, 49 days pass from felling to factory receipt, which is due to uncoordinated processes and the interrupted flow of information, with all the disadvantages that a long storage time entails [33]. The lack of internet availability in large parts of the forest areas requires offline-capable applications and slows down the just-in-time data flow. Accordingly, within the timber harvesting fleet, communication is mostly machine-to-machine (M2M). The introduction of 5G should overcome this obstacle [162]. A separate challenge is granularity, i.e., separating and assembling logs and log sections into assortments and batches [163]. Although this was only addressed to a limited extent in the paper presented here, it is still important. The challenge is posed by the unique identification of the individual log and log section in any technical and spatial manipulation [15].

**Customer demand:** The key to efficiently adapting wood-related data along the value chain is to connect the core interests of producers and those of customers. Assortments are generally not aligned with the needs of the buyer at the time of logging [33]. Not only customers and producers demand complete traceability of the raw material. Especially, government regulations promote higher demands for identification, safety and origin of products [13]. While those involved in wood supply and processing make decisions about potential cost savings and efficiency improvements, the clients can influence the market through the need for specific information about the product. This should not result in an overload of information [20].

**Economics:** Few studies analyse economic viability of transparent supply chains, as it is clear neither whether they are accurate, secure, useful and robust nor whether they are worth the investment. The reluctance to implement them clearly speaks against the viability. The heterogeneity and lack of standards that prevails at several levels is a cost factor. The many processes to be mapped in the supply chain, the many independent systems and the multitude of technical and spatial manipulations stand in the way of development, as does the low financial strength of the sector and the associated willingness to invest in the technology. The potential, on the other hand, lies in systems that generate protocols for clearly defined units in standardised data formats between all business partners in the timber supply chain at critical points in the transfer of goods [32]. The savings potential for mills in warehouse raw timber logistics alone is up to 70% [83]. Great potential also lies in improving the quality of the wood provided and the availability of currently demanded qualities in the sense of a "warehouse forest" [164]. Trust of business partners along the

value chain can be increased through transparent data availability [20]. If this trust is secured on the basis of trustworthy, mostly anonymous data storage, business partners are more willing to exploit the expected economic benefits. It can be assumed that the implementation pressure will come from the wood processing industry. In connection with this, financing from the wood industry would be conducive and driving for the process [20]. The forestry sector in general seems to be characterised by long-term investments, which may make the sector less risk-averse to invest in technology [39].

**Interoperability:** The biggest challenges are not to be found in the availability of forestry technology but in the interoperability of the resulting data and in the economic incentives to (further) develop digital technology. While harvesters and forwarders still play the most important role in collecting data, app-based data collection with smartphones and Lidar measurements are increasingly contributing to putting timber harvesting on a data-driven footing [36]. At present, it is not possible to identify the quantities of wood delivered to the mill on an individual log or log section level based on clear marking and identification methods. One resorts to random sampling [33]. Furthermore, it must be possible to transfer the already diverse data collected by the forest machines via standardised interfaces. In the best case, the collected data are already based on a common standard: a service-oriented, web-based architecture. While most authors focused primarily on the potential in planning and strategy within forestry processes, concrete guidelines for optimising the supply chain are needed [36]. There is an identification method for every process in the value chain, and the possibility of tracing is technically possible at every point in the supply chain. What is needed is an architecture into which data can flow and be exchanged among market partners in an automated way [36]. The hesitant implementation of the technology, which has already been tested and applied many times, poses a conundrum [39]. Interoperability, i.e., the ability to exchange data automatically without media discontinuity, goes hand-in-hand with agility—i.e., near real-time communication and provision of data—but also with integration—i.e., the possibility for all partners to work closely together—and finally, also with visualisation for better understanding of the processed data, e.g., via 3D forest models and maps [38]. The integration of traceability technology into the harvester head is promising. The combination of logging data and geo-coordinates allows the possibility to create a fingerprint of the log and its sections using equipment already within the harvester head. It is important that the system links the single log (parent ID) with the sections (child ID). The harvester, forwarder and truck should all have sensors installed that can recognise the identity and put new information such as the geo-coordinates on the identity [18].

The harvester determines parameters of the single tree of the exiting stand such as volume, tree species and location and is able to capture its environment by laser. In this way, the harvester could automatically transmit the relevant data to the forwarder (assortment, volume, grading and position on the skid trail). The forwarder can approach the logs more efficiently and finally transmit the position of the pile, including the relevant data, to the forestry company, which in turn can control the transport logistics in a time- and resource-efficient way. Work safety and efficiency of the personnel deployed already benefit from automated processes, remote just-in-time controlling, help with decision-making, digital work orders, hazard alerts, augmenting reality systems and much other safety-relevant and work-saving technological support that is already experiencing application in highly mechanised timber harvesting and could be introduced from other industries. In particular, the documentation of technical production and success control could no longer have to be carried out on site but could be detached from the time and place of harvesting. The same applies to logistics. The exchange of wood-related data from harvesting and wood provisioning by the transport company (assortment, localisation and quantity) without media discontinuity allows the logistics company to better organise its vehicle fleet, i.e., to realise more-efficient deployment planning and optimised route planning. This, in turn, allows for reduced storage capacities (just-in-time logistics at the forest road, intermediate storage or mill) but also the recording of other parameters, such as the quality of forest

roads, which can contribute to their speedy repair. In addition, illegal removal and sale of wood can be prevented [39].

**Distributed ledger technology:** This booster technology offers the potential to translate all processes and physical entities of the forest-timber supply chain into a digital ecosystem by allowing all partners to interact with each other. The Internet of Things becomes possible in supply chains through blockchain. Small- and medium-sized enterprises have easier access to new business areas, such as blockchain mining, which, in turn, can increase the adoption of cryptocurrencies. New specialised fields of activity are likely to mean new jobs. Overall, digitisation is likely to get a boost. Supply chains can be mapped transparently, automatically and completely and can be data-protected and have low hurdles for all partners to join. The fully implemented blockchain use-case for timber supply can make the demands on nature, people and the raw materials easier to fulfil [157]. Blockchain runs without paper-based settlements and without intermediary banks. Large stakeholders in particular but also powerful institutions such as banks and ministries are likely to have less interest in value chains being able to run completely transparently and without their control. A reliable, at best globally valid, legal framework is needed for the use of such a blockchain. The framework must also regulate the participation of cryptocurrencies, the validity of smart contracts and the intervention in transactions in case of illegality [155]. The advantage of blockchain in supply chains is its finite nature. The chain does not need to continue indefinitely and is relatively short. This saves computing power and storage space and thus energy consumption. Although blockchain technology is the talk of the town, it has not been implemented, at least in the forestry industry [163].

**Adding value:** The presented technology for tracing timber flows needs to overcome certain thresholds in order to be widely deployed and reach its full potential. These thresholds are the significant reduction of cost per cubic meter of timber; the increase in efficiency, which is mainly reflected in the utilisation of forestry machines and logistics; more transparency and trust among partners; reduction of complexity of business processes through smart digital architecture; the reduction of $CO_2$ emissions, which can be done through optimised energy yields; combating illegal logging and deforestation; and finally, the improvement of working conditions for forest managers [4,38,39]. The following challenges need to be addressed in the process towards end-to-end wood supply. The question of data protection and data ownership must be answered. Pressure from the state must be increased in two areas. First, the government must promote the use of available technology through stricter regulations, and second, it must provide the financial resources to allow the technology to achieve better market penetration [14,38,39,155]. Furthermore, forestry has special requirements for the resilience and reliability of the technology, which delays implementation from other industries. In addition, training employees is a challenge in a conservative industry that is particularly affected by demographic change [39]. Many of the methods presented, such as most forensic methods, require databases that can be reconciled and are therefore dependent on the maintenance of these databases and their data quality [3]. This can be illustrated by four dimensions, which in the best case are all mapped in a resilient traceability system: complete scope of information linked to the unit; distance of transferable data (cradle-to-gate); access of relevant actors to the data at any time at any place; and precision and reliability of the delivered data. Technologies that serve the fulfilment of all dimensions must therefore capture large amounts of data quickly and precisely and provide them with a timestamp [32].

**Merging** timber harvesting data, individual identification of timber (giving IDs and retrieving IDs) and organization of data is the main task of future technologies to trace timber. To overcome the challenges listed in the discussion, biometric fingerprint systems, as described in Section 5.4, are suitable as bridging technology, because they give the timber a unique ID that comes from the timber structure itself. On this ID, the timber-related data can be "linked". Organizing the data and linking it to the unique ID is done via data management and can succeed through IOT or blockchains. If these technologies can be

combined, it will be possible to overcome the problems of the granularity of the individual assortments and the data discontinuity tracing from the standing tree to the mill entrance.

## 7. Conclusions

We were able to present several state-of-the-art methods for identification in the timber supply chain and to locate models that digitally map the processes. Although the combination of several methods presented here is possible, we recommend the focus on a uniform, automated and worldwide standardised system. Recent scientific papers focus on the use of machine data, especially for sensing and remote sensing.

Based on the literature, we recommend:

- Use of passive optical methods: the ID of the functional unit is the internal structure of the wood or the structure of the cross-cut section, i.e., bio-metric fingerprinting.
- Use of methods that allow unique identification of the functional unit via fingerprint (RFID, DNA or biometric fingerprint).
- Installation of a data infrastructure that maps the provision and meets the needs of the customer.
- Standardised procedures and systems.
- Leverage illegal logging, carbon footprint and evidence of sustainable forest management.
- Exhaust institutional opportunities: financial aid and legislative restrictions.
- The functional unit is the selling unit—better still, the single log or log section in cubic meters over bark.
- In any spatial and technical manipulation, the identity of the wood should be recorded. The collected data should be "filed" on the wood.
- The machine data collected during harvesting, combined with coordinates, offer great potential.

Further research will be needed in the areas of biometric fingerprinting and distributed ledger technology and is recommended at this point. Improvements in sub-meter positioning, automated and standardised data transfer (M2M and D2D) and labels for the utilisation of the collected information for the end customer will promote developments towards digital mapping of the supply chain.

**Author Contributions:** Conceptualization, A.K. and T.P.; methodology, A.K.; resources, T.P.; investigation, A.K., L.S. and K.L.; writing—original draft preparation, A.K.; writing—review and editing, L.S. and K.L.; visualization, A.K.; supervision, T.P.; funding acquisition, T.P. All authors have read and agreed to the published version of the manuscript.

**Funding:** We acknowledge support by the Open Access Publication Fund of the University of Freiburg.

**Conflicts of Interest:** The authors declare no conflicts of interest.

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
