# Peer review of "Systematics of Forestry Technology for Tracing the Timber Supply Chain"

_forests, doi:10.3390/f14091718_

Round 1

Reviewer 1 Report

Authors presented a comprehensive review paper of an interesting topic supported by an extensive reference list.

I suggest following minor changes:

Optical biometric systems are identified as an optimal solution (and highlighted in the abstract as well), but discussion is focused mainly on the challenges of data handling. In my opinion, discussion should cover all reviewed technologies stressing the fact that optical biometric systems could be a solution for bridging automated identification of individual log with incorporated harvester measured data on the individual log level thus solving the problems stressed in the lines 1072-1076 and 1111-1113. In addition, the problem of recording the harvester data per traceable individual log should be elaborated in more detail in the section 5.5. Passive logical and data driven as this poses a problem when producing logs of different quality class (and value), ununiform dimensions and tree species in one felling site resulting with a large amount of ununiform assortments designated to different buyers.

Forest engineering terminology used in the section 3.2. Forestry technology and traceability in raw wood supply must be checked in detail and harmonised. For example: skidding should be used when skidders are used for roundwood extraction, and forwarding when forwarders are used. Chipping should be used instead of chopping, …  

Please state the reasons for suggesting solid cubic meter over bark as the functional unit and not the solid cubic meter under bark.

Lines 711-713  and 719-721 partial repetition; please check.

Author Response

Dear Reviewer,

thanks for taking the time commenting on my paper.
I have been able to implement your recommendations as far as possible:

  1. Optical biometric systems are identified as an optimal solution (and highlighted in the abstract as well), but discussion is focused mainly on the challenges of data handling. In my opinion, discussion should cover all reviewed technologies stressing the fact that optical biometric systems could be a solution for bridging automated identification of individual log with incorporated harvester measured data on the individual log level thus solving the problems stressed in the lines 1072-1076 and 1111-1113.

I added this information at the very end of section 5.5, as well. I think you are making a good point: Linking passive logical data from harvester and forwarder to the individual log identified by biometric fingerprinting technology ist the core statement. 

I also put another paragraph at the end of the section "discussion", following your suggestions.

2. In addition, the problem of recording the harvester data per traceable individual log should be elaborated in more detail in the section 5.5. Passive logical and data driven as this poses a problem when producing logs of different quality class (and value), ununiform dimensions and tree species in one felling site resulting with a large amount of ununiform assortments designated to different buyers.

I took your advice seriously and described some more problems in the "challenges" part of section 5.5.

3. Forest engineering terminology used in the section 3.2. Forestry technology and traceability in raw wood supply must be checked in detail and harmonised. For example: skidding should be used when skidders are used for roundwood extraction, and forwarding when forwarders are used. Chipping should be used instead of chopping, …  

I changed and harmonised the terminology.

4. Please state the reasons for suggesting solid cubic meter over bark as the functional unit and not the solid cubic meter under bark.

It is actually not relevant in this paper whether the functional unit is over bark or under bark. Thanks for the comment, which caused me to delete it.

I would still like to make a statement: The literature cites about 60% m3 over bark in similar subject spectra.  1 m3 of timber can also be used as a default unit according to ISO 215
14044 and ISO 14067. 

5. Lines 711-713  and 719-721 partial repetition; please check.

I deleted the sentence with the repetition in it.

Reviewer 2 Report

Overall: The manuscript gives a thorough overview of the state of tracking and tracing technology within the wood supply chain, with a focus on the German wood supply chain. The manuscript provides important and timely information. The manuscript could be strengthened by more clearly stating the likelihood of widespread adoption of the different technologies and providing more information on implementation costs. In addition, there are terminology and language usage issues that should be addressed prior to publication. 

Title: The title is grammatically problematic.

Line 1: One should not use the word being defined in the definition of that word.

Line 2: What if the timber is not delivered to a sawmill? What if it is delivered to a wood pellet plant?

Line 4: What is “technical and spatial manipulation?

Lines 61–62: One should not use the word being defined in the definition of that word.

Lines 72–76: These paragraphs should be combined. Avoid single-sentence paragraphs.

Line 97: Three to five sources is insufficient for a review article.

Lines 100–101: Timber harvesting and logging are synonymous.

Figure 1: Full-tree, tree-length, and cut-to-length are three different systems. Only one would be used at a time. The figure makes it seem like they are used simultaneously.

Lines 204–215: This terminology is problematic. “Delivering” generally refers to transportation to the mill, not the forest road. Forwarding only occurs within the context of a cut-to-length system. If a full-tree or tree-length system is used, then it is “skidding.” “Primary transport” refers to skidding, cable yarding, or forwarding.

Line 230: The authors stated previously that the literature review focused on the period 2011–2022, yet this paragraph leads with the period of 1998–2023.

Line 243: Forests does not use the author-date citation system.

Lines 266–267: Avoid paragraphs with only one sentence.

Line 337: “Feasibility”

Line 340: This is not a chapter.

Table 3: There is inconsistent capitalization in this table.

Line 749: Misplaced period.

There are several English usage issues that should be addressed prior to publication. I have highlighted several above; however, I primarily evaluated the manuscript content. I am not a  proofreader.

Author Response

Dear Reviewer,
thank you very much for commenting and helping me improving my writing skills. I followed your suggestions and made some changes:

Title:
I changed the title into: "Systematics of Forestry Technology for Tracing the Timber Supply Chain"

Line 1: Changed "trace" into "follow"

Line 2:  Changed it a little from "sawmill" to "mill", which shoul include factories in general. I changed it in the whole document, which made the text more consistent as well.

Line 4: I added an explanation

Line 61-62: I am sorry, but I did not find a definition.

Line 72-76:  The paragraphs are combined now.

Line 97: I understand, that this is confusing, because got much more articles. Therefore I deleted this information.

Lines 100–101: You are right. Those are synonyms. I changed it in the wohle text.

Figure 1: I did got your point. The figure describes not the systems of full-tree, tree-length, and cut-to-length, but the condition of the tree itself. The process begins with the full tree, it is cut in tree length and after that it is cut into assortments.

Lines 204–215: You are right. I followed your suggestion and changed it. This helped a lot.

Line 230: I changed it, following your suggestion, in the method section. Some older articles were still relevant and it was decided to use them.

Line 243: I changed it in the whole article

Lines 266–267: solved

Line 337: solved

Line 340: solved

Table 3: I changed it, which led to more consistency

Line 749: Done

Thanks again. This one is my first article. Therefore I appreciate your effort. It led to an improvement of the text itself and propably my writing skills, as well.